# Isolation of intact extracellular vesicles from cryopreserved samples

**Shannon N. Tessier**[1,2], **Lauren D. Bookstaver**[1], **Cindy Angpraseuth**[1], **Cleo J. Stannard**[1], **Beatriz Marques**[1], **Uyen K. Ho**[1,3], **Alona Muzikansky**[4], **Berent Aldikacti**[1,3¤a], **Eduardo Reátegui**[1,3¤b], **Daniel C. Rabe**[1,3], **Mehmet Toner**[1,2], **Shannon L. Stott**[1,3]*

**1** Department of Surgery, Center for Engineering in Medicine and BioMEMS Resource Center Massachusetts General Hospital, Harvard Medical School, Boston, MA, United States of America, **2** Shriners Hospitals for Children—Boston, Boston, MA, United States of America, **3** Department of Medicine and Cancer Center, Massachusetts General Hospital, Harvard Medical School, Charlestown, MA, United States of America, **4** Biostatistics Center, Massachusetts General Hospital, Boston, MA, United States of America

¤a Current address: Department of Biochemistry and Molecular Biology, Molecular and Cellular Biology Program, University of Massachusetts Amherst, Amherst, MA, United States of America
¤b Current address: William G. Lowrie Department of Chemical and Biomolecular Engineering, Comprehensive Cancer Center, The Ohio State University, Columbus, OH, United States of America
* SSTOTT@mgh.harvard.edu

**Data Availability Statement:** All relevant data are within the paper and its Supporting Information files. Reasonable requests for other raw data supporting this work can be made to Dr. Stott.

## Abstract

Extracellular vesicles (EVs) have emerged as promising candidates in biomarker discovery and diagnostics. Protected by the lipid bilayer, the molecular content of EVs in diverse biofluids are protected from RNases and proteases in the surrounding environment that may rapidly degrade targets of interests. Nonetheless, cryopreservation of EV-containing samples to -80˚C may expose the lipid bilayer to physical and biological stressors which may result in cryoinjury and contribute to changes in EV yield, function, or molecular cargo. In the present work, we systematically evaluate the effect of cryopreservation at -80˚C for a relatively short duration of storage (up to 12 days) on plasma- and media-derived EV particle count and/or RNA yield/quality, as compared to paired fresh controls. On average, we found that the plasma-derived EV concentration of stored samples decreased to 23% of fresh samples. Further, this significant decrease in EV particle count was matched with a corresponding significant decrease in RNA yield whereby plasma-derived stored samples contained only 47–52% of the total RNA from fresh samples, depending on the extraction method used. Similarly, media-derived EVs showed a statistically significant decrease in RNA yield whereby stored samples were 58% of the total RNA from fresh samples. In contrast, we did not obtain clear evidence of decreased RNA quality through analysis of RNA traces. These results suggest that samples stored for up to 12 days can indeed produce high-quality RNA; however, we note that when directly comparing fresh versus cryopreserved samples without cryoprotective agents there are significant losses in total RNA. Finally, we demonstrate that the addition of the commonly used cryoprotectant agent, DMSO, alongside greater control of the rate of cooling/warming, can rescue EVs from damaging ice formation and improve RNA yield.

**Funding:** This work was supported by the Wang Family Foundation Grant (SLS), funding from the NIH/NCI (1R01CA226871-01A1, SLS), the American Cancer Society (132030-RSG-18-108-01-TBG, SLS) and the Massachusetts General Hospital Claflin Distinguished Scholar Award (SLS). SNT held a Career Development Award from the American Heart Association (18CDA34110049) and K99/R00 Pathway to Independence Award (K99/R00 HL143149). SNT also greatly acknowledges funding from the MGH Department of Surgery Eleanor and Miles Shore Fellowship and the MGH Claflin Distinguished Scholar Award. We also note that this material is based upon work supported, in part, by the National Science Foundation under Grant No. EEC 1941543.

**Competing interests:** Drs. Stott, Tessier, and Toner are inventors on several provisional patents on the topic of blood stabilization and cryopreservation of EVs, cells, tissues, and organs (W02016022433A1, W02018005802A1, W02020163774A1). These do not alter our adherence to PLOS ONE policies on sharing data and materials. There are no restrictions on sharing of data and/or materials as a result of these competing interests. Researcher's interests are managed by the MGH and Partners HealthCare in accordance with their conflict-of-interest policies.

# Introduction

Extracellular vesicles (EVs), including microvesicles and exosomes, have recently attracted considerable attention for their potential in biomarker discovery and diagnostics. EVs carry diverse cell cargo including lipids, proteins, metabolites, DNA, mRNAs, microRNAs, and other non-coding RNAs. This cell cargo is strategically packaged and released from the parent cell and can travel long distances before delivery to recipient cells, completely changing the landscape of cell-to-cell communication. EVs have been shown to actively participate in physiological and pathological processes [1], and the number of EV-mediated roles continuously increase. For example, tumor EVs have been shown to promote angiogenesis, invasiveness, and metastasis [2–4], while some of the earliest characterization of EVs showed secretion during reticulocyte maturation removes obsolete membranes and proteins during "reverse endocytosis" [5,6].

EVs are particularly attractive as a diagnostic since they are abundant in circulation. Thus, EVs provide a unique opportunity for non-invasive, continuous sampling to discover a range of information about the whole body, as compared to invasive surgical biopsies which only sample a small region of the body. While blood is highly susceptible to rapid deterioration *ex vivo* [7,8], the intracellular contents of EVs are relatively protected from extracellular RNAses and proteases. This is due to their protective lipid bilayer and increased stability of miRNAs [2]. For example, labile molecules such as unprotected circulating RNAs have been shown to degrade in less than 3 hours in plasma [9] and this degradation can eliminate the ability to detect more than 99% of transcripts of interest [10]. Of course, this labile nature would be detrimental for diagnostics and dissemination purposes, especially since RNA has been shown to hold clinically relevant and actionable information [11].

Despite the promise of EVs and their molecular cargo for diagnostics, unfortunately, there are no universally applied/implemented processing and storage conditions, despite efforts to standardize [12,13]. While there is some consensus that fresh samples are preferred [14], this is often impracticable if not impossible to achieve and laboratories have developed their independent strategies likely based on the specific needs of their research goals. In general, processing of peripheral blood or cell culture supernatant for downstream molecular analysis of EVs involves room temperature isolation followed by sample storage, with large variations in processing steps. For example, the choice of anticoagulants, the time frame between sample collection and processing, or the isolation method have been shown to influence EV yields and biomarkers from blood [13,15,16]. Of particular relevance to the present work, EVs are also influenced by the method of storage (reviewed in [15]). Indeed, the field of cryobiology has comprehensively documented that poor sample handling and preservation practices may lead to severe degradation of bioanalytes thereby compromising the quality of biological and diagnostic information obtained [17]. Thus, adequately addressing the impacts of preservation practices on EVs and their cargo is paramount.

Select papers have already begun exploration of the stability of EVs and their molecular signature as a function of storage, including both short-term stability and long-term banking. For relatively short-term storage durations, researchers suggest EVs from conditioned media and bodily fluids can be maintained for up to a week or 5 days, respectively, at 4°C [18], whereas others have shown that EV diameter significantly changes within 2 days and this is influenced by the storage temperature (between 4–37°C) [19]. For longer-term cryopreservation of EVs (i.e. ≤-80°C), one study showed banked samples can still yield high quality RNA after up to 12 years of storage [2,20], although no fresh controls are presented to ascertain how the stored samples may have changed over time. Others have shown that cryopreservation of exosomes in serum-free media at -80°C showed more heterogeneous

shapes with scanning electron microscopy (SEM), as compared to fresh exosomes, and provided evidence of RNA degradation as a function of storage durations ranging from 9 days to 2 years [21]. However, it should be noted that these cryopreserved samples still contained a variety of RNA molecules characteristic of fresh samples, suggesting specific subsets of RNA may not be selectively degraded. Further, the authors showed that through the addition of the cryoprotectant agent (CPA), DMSO, the sizes and shapes of a certain percentage of exosomes were adequately cryopreserved; however, RNA nonetheless appeared degraded in 2-year-old banked samples [21]. There is some agreement that multiple freeze-thaw cycles can affect EVs [15], although others have reported EVs are relatively insensitive to freeze/thaw cycles [19,22]. Finally, researchers have shown that the thawing conditions play a role in EV recovery whereby samples thawed on ice showed a lower recovery than those thawed at room temperature or 37˚C [23]. Taken together, these studies clearly suggest additional work is needed to identify storage and handling methods which are optimal for retention of EV yield, function, and molecular content. More specifically, we aimed to add to this body of literature by comparing frozen/stored samples to matched, fresh controls obtained from plasma and cell culture supernatants. We also aimed to evaluate the effect of different isolation methods, including ultracentrifugation and qEV Columns, and to determine the impact of freezing/thawing versus banking in the presence of cryoprotectant agents. Further, it is possible multiple preservation methods will be required to address specific needs–for example, perhaps samples can be rapidly transferred to the lab for processing within a relatively short time window (on the order of days) versus situations where true long-term banking is required so samples can be collected over years.

While the lipid bilayer of EVs does act as a protective layer in circulation against degradative enzymes, it is well known and broadly accepted that lipid bilayers are sensitive to low temperatures [24–28]. As temperatures drop, the sample undergoes dramatic phase changes as more and more water is trapped as ice. Further, the process of traversing the intermediate temperature zone between -15 and -60˚C is often thought to be a major source of damage [29], in addition to the potential for degradation of analytes over time. Ultimately, a mixture of physical and biological factors contributes to damage of biologics during cryopreservation, each of which can contribute to "cryoinjury." Without proper prevention, these factors can have serious downstream effects. Despite well-known mechanisms of damage to cells during cryopreservation, EVs are very different biological systems necessitating comprehensive studies to find the right conditions for storage. For example, EVs are much smaller and thus the dynamics of intra-EV ice formation and water/solute balance will be different from cells. Yet, since EVs are composed of lipid bilayers which have shared properties with cells and since evidence has shown that lipid bilayers can catalyze ice formation [30,31], EVs are likely not innocuous to freezing-induced damage.

Here we systematically evaluate EV particle counts and RNA yield/quality by directly comparing matched fresh versus stored samples containing EVs. The primary purpose is to assess the effect of freezing/thawing down to -80˚C and relatively short-term cryopreservation durations (up to 12 days) on EV and RNA yield/quality. Our results suggest that frozen samples can indeed produce high-quality RNA; however, we note that when comparing matched fresh versus cryopreserved samples without cryoprotective agents there are significant losses in total RNA, with one mechanism of loss due to freeze-thaw cycles. Importantly, we describe a simple protocol which involves the addition of the commonly used cryoprotectant agent, DMSO, to rescue EVs from damaging ice formation and improve RNA yield. It should be emphasized that the present study does not address changes in RNA as a function of time. Thus, additional

research is required to ascertain the impact of cryopreservation approaches more comprehensively on the molecular cargo of EVs.

## Materials and methods

### Ethics statement

All plasma obtained for our study was drawn from healthy individuals at Mass General Hospital, following written consent under our institution approved IRB protocol (2009-P-000295).

### Blood collection

All healthy donors were not taking medications at the time of blood draw. Plasma samples were drawn into a BD Vacutainer PPT tube. Immediately after collection of blood, the tube was inverted 8–10 times and centrifuged in a swing-out rotor at room temperature (1,100 rcf for 10 minutes). In a sterile cell culture hood, the plasma layer was aspirated into a 10 mL syringe using a 16G needle and subsequently filtered with a 0.8 μm filter. All plasma samples were collected from healthy donors at MGH thus were processed within two hours of blood draw (e.g. no shipping or transport time), as described further below.

### Cell culture and media collection

BM1 palm-tdTomato cells were grown in high-glucose DMEM with L-glutamine (Corning) supplemented with 10% fetal bovine serum (Gibco) and 100 U/ml penicillin and 100 ug/ml streptomycin (Gibco). BM1 cells are a highly invasive bone-metastatic variant of MDA-MB-231 cells (the most used cell line to study triple negative breast cancer) that generate a lot of EVs. To generate fluorescent EV reporters for direct visualization of cargo, a palmitoylation signal was genetically fused to the N-terminus of tdTomato [32,33]. Cells were grown to 90% confluence then washed three times with phosphate buffered saline without calcium and magnesium (Corning). After washing, cells were placed in serum free DMEM for 24 hours to collect secreted EVs.

### EV sample processing

EVs were isolated using the handling protocol illustrated in Fig 1. Samples were immediately aliquoted into fresh versus stored experimental conditions (equal volumes of 4 mL per experimental condition). Fresh samples were then processed either by ultracentrifugation or using qEV Columns (Izon Science, Cat# qEVoriginal / 70 nm), followed by either particle analysis or RNA extraction (RNA concentration ranged from 17–1300 pg/μL), please see corresponding sections below that elaborate on each of these methods (i.e. *EV isolation*, *EV quantification*, and *RNA isolation and characterization*). Stored samples were placed into a -80°C freezer and thawed 10–12 days later. After thaw, EVs were isolated by ultracentrifugation or qEV columns, and immediately processed by either particle analysis or RNA extraction. We also tested two different cooling and warming protocols. For cooling, samples were either cryopreserved by adding cryogenic vials directly to the -80°C freezer versus first placing in a Nalgene freezing container where they were held for 6 hours at -80°C (Mr. Frosty, Cat#C1562). For thawing, samples were either removed from the -80°C freezer and thawed at room temperature (~1–2 hrs to thaw) or placed in a warm water bath (37°C, ~1–3 mins to thaw) until only a sliver of ice remained in the cryogenic vial. Moreover, two storage durations were tested (i.e. 10–12 days versus hours) to test if RNA loss was due to the freezing or storage duration. Finally, we also tested freezing samples in the presence of the well-known cryoprotectant agent dimethyl

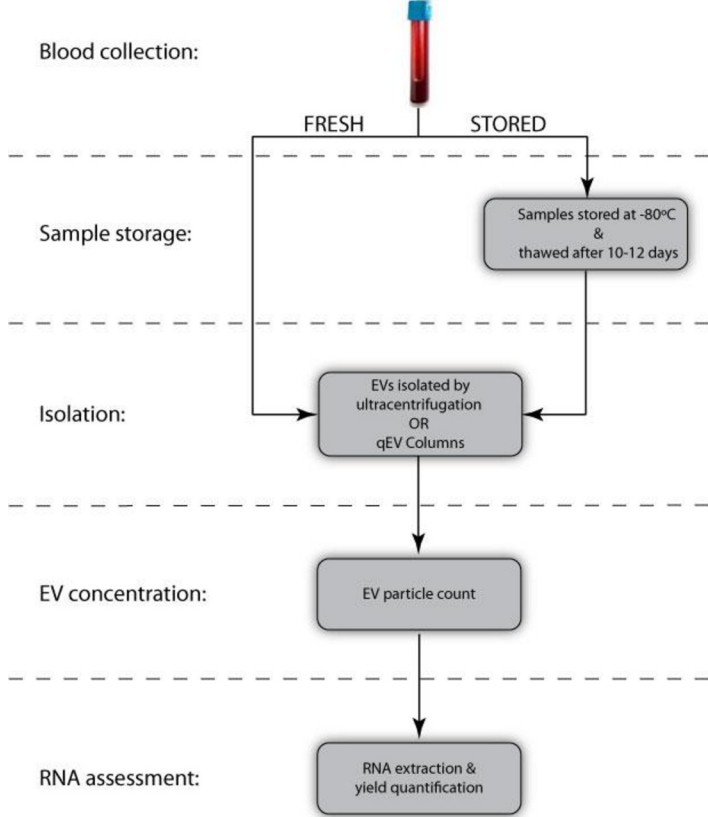

**Fig 1. Sample handling protocol for EV isolation, particle analysis, and RNA characterization.** Fresh versus stored samples containing plasma- or media-derived EVs were assessed for particle count (i.e. yield) and/or RNA yield/ quality after EV isolation using ultracentrifugation or qEV columns.

sulfoxide (variously called DMSO, Me$_2$SO, (CH$_3$)$_2$SO). Prior to cryopreservation, 10% v/v DMSO was combined with plasma samples and mixed by pipetting up and down.

## EV isolation

Plasma-derived EVs were isolated using a differential ultracentrifugation method, as others have also used [34]. Samples were first spun at 300 rcf for 10 minutes at 4˚C to remove cell contamination. Then, the supernatant was centrifuged at 2000 rcf for 10 minutes at 4˚C to remove cellular debris and large vesicles. Subsequently, the supernatant was collected and filtered through a 0.8µm filter. For ultracentrifugation, the supernatants were spun at 4˚C with a Ti80 rotor for 90 minutes at 100,000 rcf (L8-80M Ultracentrifuge, Beckman Coulter, Brea, CA). Then the supernatant was very carefully aspirated without disturbing the pellet before resuspending the pellet in double 0.22µm filtered PBS. Media-derived EVs were isolated using qEV Columns since these are expected to be devoid of contaminating proteins or other particles below 70 µm, including lipoprotein particles and RNA containing ribonucleoprotein complexes. Briefly, media was spun at 2000 rcf for 10 min at room temperature to remove larger particles, dead cells, and debris. Then, 15 mL of media was concentrated using Amicon Ultra-15 10 kDa filters (Millipore/Sigma) at 4000 rcf at 4˚C for approximately 25 minutes down to a volume of 500 µl. Concentrated media was then loaded onto 70 nM qEV columns, and 500 µl fractions 7–9 were collected for further EV analysis.

## EV quantification

EVs were quantified using a tunable resistive pulse sensing (TRPS) qNano instrument (Izon Science, New Zealand) and NP300 pore size membrane. Firstly, the upper and lower fluid cells were primed with PBS and using calibration beads (qNano CPC400E) at three pressures (5, 10, and 15 mbar) by a water-based variable pressure module. Similarly, EV concentration and size distribution for control and experimental samples were quantified by adding the sample to the upper fluid cell and taking readings at three pressures. Data were acquired and analyzed using the Control Suite Software (Izon Science, version 3.3.2.2001). Particle rate is proportional to particle concentration and the applied pressure. We compared values across experimental conditions with a constant pressure hence values presented are proportional to particle concentration.

## RNA isolation and characterization

RNA was extracted using either the Qiagen RNeasy Plus Micro Kit (cat#74034) or miRNeasy Mini Kit (cat#217004) to ensure the trends we observed were not specific to downstream handling. This is especially relevant since miRNeasy spin columns have a higher binding affinity for small RNAs, as compared to RNeasy kits and the presence or absence of small RNAs can impact RIN values. Briefly, Trizol LS reagent was added to the sample 3:1 v:v before vortexing at room temperature and adding 1:5 v:v of chloroform. After vortexing and a brief incubation period at room temperature, samples were centrifuged at 12,000 rcf for 15 minutes at 4°C and the aqueous phase was transferred to a new tube. Next, 100% ethanol was added (1.5× the volume of the aqueous phase) before transferring to either RNeasy or miRNeasy spin columns and proceeding as per manufacturer's protocol. RNA concentrations and quality were determined using the Bioanalyzer (Agilent Technologies, Agilent RNA Pico Kit, Cat#5067–1513), as per standard protocols. The Bioanalyzer uses capillary electrophoresis to assess RNA sizing, quantity, and integrity prior to downstream analysis. A fluorescent dye molecule intercalates into the RNA strand and bands are detected by their fluorescence. Bands are translated into both a gel-like image and peaks in the electropherogram. Each peak in the electropherogram is quantified in fluorescence units (y-axis) as a function of time (x-axis) since the speed in which RNA fragments move through the gel is proportional to their size. As such, RNA peaks closer to 20 seconds are smaller as compared to those that approach 70 seconds. The RNA Integrity Number (RIN) is a Bioanalyzer algorithm for assigning integrity values to RNA.

## RNase treatment

To ensure decreases in RNA yield were due to loss of intra-EV RNA, we treated EVs isolated using ultracentrifugation or qEV Columns with RNase A. Briefly, EVs resuspended in PBS were incubated at 37°C for 30 minutes in the presence of 8.25 μg/ml of RNase A (Thermo Scientific, EN0531). RNase inhibitor was then added to samples and incubated at 37°C for 10 min to inactivate RNase A.

## Immunoblot analysis

To ensure our EV preparations using both ultracentrifugation and qEV Columns did produce pure EVs, we performed immunoblot analysis against CD9. Briefly, samples were lysed in RIPA buffer, sonicated, then spun at 10,000 rcf at 4°C for 10 minutes to remove debris. Protein levels were quantitated using a BCA (Pierce). Samples were boiled for 5 min at 95°C after the addition of 4× NuPAGE sample buffer (Invitrogen, NP0007). Approximately 50 μg of protein was loaded onto a 4–20% TGX Gel (Bio-Rad) and run at 80 V. Samples were then transferred

to a PVDF-LF membrane using a wet transfer method at 80 V for 1 hour. Samples were blocked in 5% milk in TBS-T and blotted for CD9 (Bio-Rad, MCA469) and imaged using a Li-Cor Odyssey scanner.

## Quantification and statistics

Each biological replicate from plasma represents one individual whereby the same blood draw from the same day was divided into paired fresh and stored samples of equal volume. Similarly, biological replicates for media samples are taken from the same cell culture plate from the same day and divided into paired fresh and stored samples of equal volumes. In all figures, solid lines between data points link the corresponding fresh and stored samples from a biological replicate. Absolute values are graphed as box-and-whiskers plots showing median, inter-quartile range, maxima/minima, and all individual data points (n = 4–11). Statistical analysis used a one-sided Wilcoxon signed-rank test. All tests were performed with GraphPad Prism 7 (version 7.04).

## Results

### Impact of cryopreservation on particle count and size distribution of plasma-derived EVs isolated using ultracentrifugation

We assessed the particle count of plasma samples using the handling protocol, as described in Fig 1, to emulate the most common banking practices. Samples were analyzed for particle concentration and size distribution comparing freshly processed versus stored samples (-80°C storage for 10–12 days), as illustrated in Fig 2. On average, the plasma-derived EV concentration of stored samples decreased significantly as compared to fresh, non-frozen samples whereby stored samples were $1.25 \times 10^9 \pm 9.68 \times 10^8$ EVs/mL and fresh samples were $5.38 \times 10^9 \pm 2.60 \times 10^9$ EVs/mL (p = 0.031, n = 5; Fig 2A). As depicted in S1A–S1D Fig, this results in a much lower particle rate (i.e. the rate at which particles are counted in the qNano instrument) of stored versus fresh samples. Further, we assessed the average particle size and detected no significant difference between fresh and stored plasma-derived EVs for the size range evaluated (p = 0.31, Fig 2B). This is also reflected in Fig 2C and 2D which shows a representative concentration versus particle diameter histogram, with concentration of fresh versus stored EVs shown as dark grey and red bars, respectively. Histograms showing concentration of EVs as a function of size for additional donors are shown in S2A and S2B Fig.

### Impact of cryopreservation on RNA yield of plasma-derived EVs isolated using ultracentrifugation

All samples followed the handling protocols, as described in Fig 1, and used ultracentrifugation to isolate EVs. We assessed the RNA from plasma-derived EVs using two routine RNA extraction kits (i.e. Qiagen RNeasy shown in Fig 3A versus miRNeasy kit shown in Fig 3B). Using the RNeasy RNA extraction kit, Fig 3A shows the RNA yield of stored samples decreased significantly to 46.7% of fresh samples (p = 0.011, n = 10, average of fresh samples was 45.1 ± 24.2 pg/μl and stored samples was 21.0 ± 18.6 pg/μl). S3A Fig shows corresponding RIN values of stored samples decrease significantly to 59.1% of fresh controls (p = 0.0039, average of fresh samples was 6.95 ± 1.00 and stored samples was 4.11 ± 2.61). S3B and S3C Fig show representative electropherograms obtained from the Bioanalyzer whereby a clear loss in RNA yield is observed by overlaying fresh (black line) versus stored (red line) samples. Similarly, RNA yield also decreased significantly when extraction used the miRNeasy spin columns, whereby values of stored samples were only 52.3% of freshly processed samples (p = 0.014, n = 9, average of

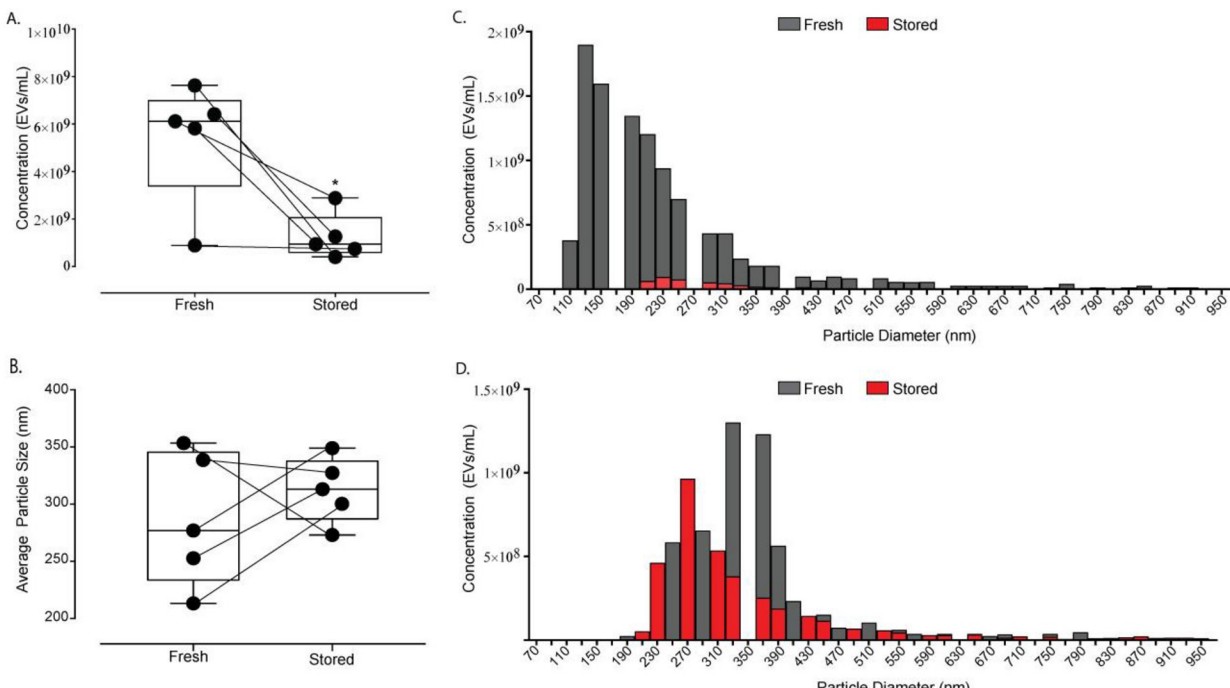

**Fig 2. Impact of cryopreservation on particle count and size distribution of plasma-derived EVs.** All plasma samples were added directly to the -80°C freezer (without a freezing container), thawed at room temperature, and isolated using ultracentrifugation. (A) Concentration of EVs in fresh and stored samples are expressed as box-and-whiskers plots showing median, interquartile range, maxima/minima, and all individual data points (n = 5). All individual data points (or biological replicates) consist of paired fresh and stored samples that were obtained from the same individual in one blood draw. Solid lines between data points link the corresponding fresh and stored samples from a given individual. (B) Average particle size (nm) of EVs in stored versus fresh samples are expressed as box-and-whiskers plots showing median, interquartile range, maxima/minima, and all individual data points (n = 5). (C-D) Representative histogram showing EV concentration binned by particle diameter (nm) with fresh samples shown in dark grey and stored samples in red. Each histogram shows data from one biological replicate and other biological replicates are shown in S2 Fig. Data were analyzed using the Wilcoxon signed-rank test (p<0.05); an asterisk indicates statistical significance.

fresh samples was 602.8 ± 464.7 pg/µl and frozen samples was 315.5 ± 394.1 pg/µl; Fig 3B). Finally, the RIN value also decreased significantly with stored values 82.9% of fresh samples (p = 0.031, average of fresh samples was 2.93 ± 0.85 and stored samples was 2.43 ± 0.44; S3D Fig). Also shown in S3E and S3F Fig are representative electropherograms obtained from the Bioanalyzer (fresh versus stored denoted as a black or red line, respectively).

## Impact of cryopreservation on RNA yield of media-derived EVs isolated using qEV Columns

All samples followed the handling protocols, as described in Fig 1, although EVs were isolated using qEV Columns (Izon Sciience). Fig 3C shows a statistically significant decrease in RNA yield, whereby stored samples were 58.4% of fresh samples (p = 0.024, n = 10, average of fresh samples was 46.5 ± 34.1 pg/µl and stored samples was 27.2 ± 23.5 pg/µl).

## Impact of freeze-thaw on RNA yield of plasma-derived EVs

All RNA from plasma-derived EVs were extracted using RNeasy spin columns, isolated by ultracentrifugation, and followed the handling protocols, as described in Fig 1. However, plasma samples were first placed in a freezing container (see Methods for details) that is designed to achieve a rate of cooling close to -1°C/min, before placing samples in the -80°C. Further, samples were left for only 6 hours (which is enough time for the samples to equilibrate

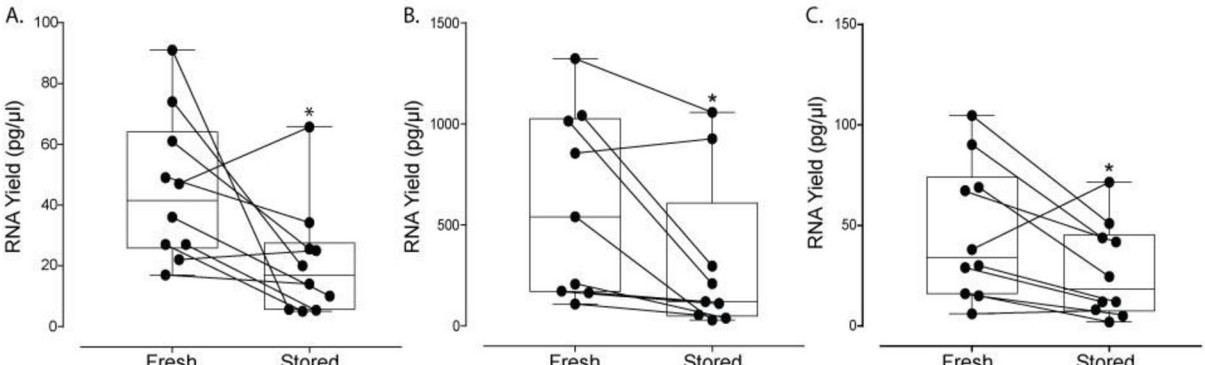

**Fig 3. Impact of cryopreservation on RNA yield of plasma- and media-derived EVs.** All plasma and media samples were added directly to the -80°C freezer (without a freezing container) and thawed at room temperature. EVs from plasma were isolated using ultracentrifugation and EVs from media were isolated using qEV Columns (Izon Science). All individual data points (or biological replicates) consist of paired fresh and stored samples that were obtained from either the same individual in one blood draw (plasma-derived EVs) or same cell culture plate (media-derived EVs). Solid lines between data points link the corresponding fresh and stored samples for a given biological replicate. (A) RNA yield of plasma-derived EVs in fresh versus stored samples isolated using ultracentrifugation and extracted using the RNeasy spin columns are expressed as box-and-whiskers plots showing median, interquartile range, maxima/minima, and all individual data points (n = 10). (B) RNA yield of plasma-derived EVs in fresh versus stored samples isolated using ultracentrifugation and extracted using the miRNeasy spin columns are expressed as box-and-whiskers plots showing median, interquartile range, maxima/minima, and all individual data points (n = 9). (C) RNA yield of media-derived EVs in fresh versus stored samples isolated using qEV Columns and extracted using the miRNeasy spin columns are expressed as box-and-whiskers plots showing median, interquartile range, maxima/minima, and all individual data points (n = 10). Data were analyzed using the Wilcoxon signed-rank test (p<0.05); an asterisk indicates statistical significance.

to -80°C) before thawing in a warm water bath (Fig 4). This was done to address two central questions. Firstly, we assessed if the process of freezing itself resulted in loss of RNA yield or if it was the storage duration of 10–12 days, as used in all previous experimental conditions thus far. Secondly, we wondered if simply controlling the cooling rate and rapid warming to pass through the dangerous intermediate temperature zone between -15 and -60°C, that is known to cause damage, could help mitigate loss of RNA yield. However, these simple additions to the freezing/thawing protocol were not enough to fully rescue RNA yield whereby samples frozen with a freezing container and rapidly thawed still showed a statistically significant decrease in RNA yield, as compared to fresh samples (p = 0.039, n = 7, average of fresh samples was

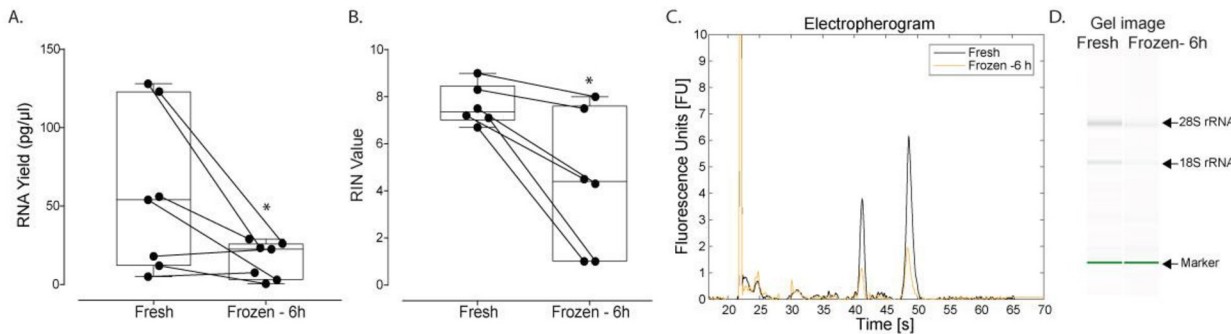

**Fig 4. Impact of freeze-thaw on RNA yield of plasma-derived EVs.** In all cases, cryovials were first placed in a Nalgene freezing container before adding to the -80°C freezer and thawed in a warm water bath. RNA was extracted using RNeasy spin columns. (A) RNA yield and (B) RIN of plasma-derived EVs in frozen/thawed (samples were held at -80°C for 6 hours) versus control (fresh) samples are expressed as box-and-whiskers plots showing median, interquartile range, maxima/minima, and all individual data points (n = 7). (C) A representative electropherogram and (D) gel image showing RNA traces obtained from the Bioanalyzer. Data were analyzed using the Wilcoxon signed-rank test (p<0.05); an asterisk indicates statistical significance. Fresh (black) versus Frozen (yellow) denotes control (not frozen) versus samples frozen and thawed with a freezing bucket and rapidly warmed in a water bath, respectively.

56.6 ± 51.1 pg/µl and frozen samples was 16.0 ± 11.9 pg/µl; Fig 4A). Similarly, there was a significant difference in RIN values comparing control and frozen samples (p = 0.016, average of fresh samples was 7.63 ± 0.86 and frozen samples was 4.38 ± 3.02; Fig 4B).

### Impact of cryoprotectant agents on RNA yield of plasma-derived EVs

All RNA from plasma-derived EVs were extracted using miRNeasy spin columns, used ultracentrifugation to isolate EVs, and followed the handling protocols, as described in Fig 1. Further, all samples were frozen using a freezing container and thawed rapidly in a warm water bath. When samples were frozen with DMSO there was no statistically significant difference between freshly processed and stored samples with respect to RNA yield (p = 0.42, n = 6; Fig 5A) and RIN values (p = 0.42; Fig 5B). A representative electropherogram and gel image obtained from the Bioanalyzer shows traces are very similar when comparing fresh (black line) versus samples stored with DMSO (green line; Fig 5C and 5D).

### RNase treatment and immunoblots

We treated EVs isolated by both ultracentrifugation and qEV Columns with RNase A to ensure loss of EVs during cryopreservation was due to intra-EV RNA (S4 Fig). There was no statistically significant difference between RNase A and control samples for either EVs isolated from plasma using ultracentrifugation (p = 0.24, n = 6; S4A Fig) or media using qEV Columns (p = 0.4857, n = 4; S4B Fig). Finally, we performed immunoblot analysis on EVs collected using both of our isolation methods to ensure our preparations did in fact contain pure fractions of EVs. S4C Fig shows positive staining for CD9 for both EVs isolated from plasma and media.

## Discussion

In the past decade, there has been increasing interest in the isolation and characterization of EVs and their contents for biomarker discovery in broad fields of medicine. Due to the protective lipid bilayer, bioanalytes contained within EVs are relatively more stable than unprotected RNA in plasma [9]. However, systematic studies evaluating the direct effect of cryopreservation protocols on EVs and the signatures they carry has not been completely described. Yet,

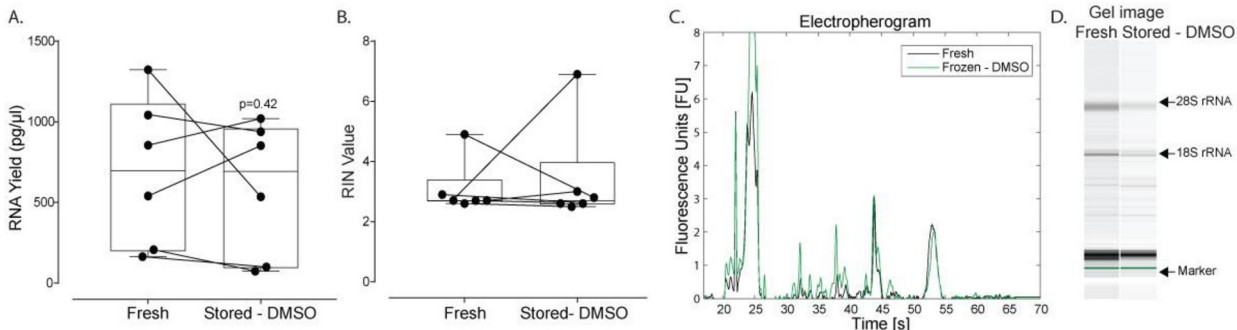

**Fig 5. Impact of cryoprotectant agents on RNA yield of plasma-derived EVs.** In all cases, cryovials were first placed in a freezing container before adding to the -80˚C freezer and thawed in a warm water bath. RNA was extracted using miRNeasy spin columns. (A) RNA yield and (B) RIN of plasma-derived EVs in stored samples (10–12 days) supplemented with DMSO (10% v/v) versus fresh samples are expressed as box-and-whiskers plots showing median, interquartile range, maxima/minima, and all individual data points (n = 6). (C) A representative electropherogram and (D) gel image showing RNA traces obtained from the Bioanalyzer. Data were analyzed using the Wilcoxon signed-rank test (p<0.05); an asterisk indicates statistical significance. Fresh (black) versus Stored—DMSO (green) denotes freshly processed versus samples stored in the presence of cryoprotectant agents, respectively.

this research topic warrants further investigation since the impacts on RNA yield and quality can influence downstream assays. Indeed, the clinical state of the patient can only be accurately assessed when the specimen is in an optimal condition required for the analysis. In this study, we build on previous work to describe the impact of freezing plasma and media at -80°C for isolation and RNA extraction from EVs. Further, we describe how the addition of the cryoprotectant agent, DMSO, can overcome some forms of cryoinjury, although longer-term preservation studies are a necessary next step. Importantly, this method is relatively easy, does not require any expensive equipment, and is compatible with downstream assays since DMSO should be removed during isolation, making it feasible for research and clinical implementation.

There are several well-known challenges associated with cryopreservation of cells in suspension [26,26,35–37]; however, the effects on EVs are significantly less understood. In the present study, we evaluated the impact of a relatively short storage duration (10–12 days) at -80°C on EVs in plasma and media, with matched fresh controls obtained from the same donor/plate. We found a statically significant decrease in the relative concentration of plasma-derived EVs (Fig 2A), suggesting that there is a loss in total EVs. This decrease in total EV count is also reflected in significant decreases in RNA yield from plasma-derived EVs isolated by ultracentrifugation and media-derived EVs isolated using qEV Columns, as observed in Fig 3A–3C. In addition, we sought to further understand the mechanism of plasma-derived EV loss since we could not delineate if decreases in EV concentration are due to the storage duration, the actual process of freezing and thawing, or both. When we exposed EVs to only a short freeze-thaw cycle (Fig 4), we nonetheless observed a decrease in the overall RNA yield, suggesting freezing/thawing induced damage which contributes to a loss in EV concentration. This data is in agreement with some published work which suggests multiple freeze-thaw cycles can affect EVs [15,38]; however, others have shown multiple freeze-thaw cycles did not influence microvesicle counts in plasma after freezing [22]. While the exact reason for these apparent contradictory results are not clear, differences in sample processing, particularly in the method used for particle count, may be responsible. It should also be noted, however, that this does not suggest storage duration does not have an impact on EVs and their contents since storage durations on the order of months or years may be required to adequately address this question.

While there is an overall loss in EV particle count, there was no change in the average particle size of plasma-derived EVs between 190–500 nm (Fig 2B). This suggests the loss may be unspecific, at least we did not observe an obvious trend in the size of EVs dictating susceptibility to cryo-mediated lysis in the specific size range measured in the present study (i.e. ~190–500 nm). A similar conclusion can also be inferred from the electropherograms presented in S3B, S3C, S3E and S3F Fig whereby RNA was extracted from plasma-derived EVs using both the RNeasy and miRNeasy extraction kits. While the electropherograms clearly show a loss in overall RNA, cryopreserved samples still contain a variety of RNA which closely parallel those obtained from fresh controls, regardless of the extraction method. Since the RNeasy and miRNeasy prefer larger versus smaller RNA, respectively, the comparison between these extraction methods tells us that it is not a preferential loss of larger RNAs, but likely an unbiased loss of RNA. This is in agreement with other studies which have shown RNA isolated from frozen exosomes for 9 days and 2 months still contained a variety of RNA characteristic of fresh samples [21]. It should be noted, however, that these observations are inferred from electropherogram traces without direct quantification of the subtypes of RNA and thus should be interpreted with great caution.

In addition to RNA yield, we also measured the RNA integrity number (RIN) which showed less clear trends, at least after short preservation durations of up to 12 days. It should be emphasized that RIN values are calculated based on degradation of rRNA bands and EVs

vary in the composition RNA cargo. Importantly, we compare all RIN values against their freshly isolated control from the same individual to account for these variations. Nonetheless, while RIN can provide some complimentary information about RNA quality in EVs it should be interpreted within the context of the study. S3A and S3D Fig shows that cryopreserved plasma samples showed a statistically significant decrease in the relative RIN value; however, direct observation of the associated electropherogram show no clear indication of a decrease in RNA quality (as evidenced by a clear shift in the amount of RNA in the slow, as compared to the fast region). Thus, we suggest that despite the impact of cryopreservation on total EVs and RNA yield, stored samples should nonetheless yield high quality RNA, as others have also suggested [2,20]. However, others have also shown that during *extended preservation*, samples banked for 2 years resulted in loss of 18S and shifting of RNA towards smaller nucleotides [21], suggesting that RNA degradation may simply not be as dramatic in our relatively short-term stored samples of up to 12 days.

With evidence to suggest losses in RNA yield because of cryopreservation, we aimed to describe a simple solution to improve yields, especially for samples with low number of EVs where loss could be detrimental. Freezing during cryopreservation results in the formation of an ice fraction and non-frozen fraction. This non-frozen fraction can transition into an injurious and damaging crystalline state, especially during long-term cryopreservation [39]. To prevent this, cryopreservation protocols generally aim to store samples below the glass transition temperature to enable this non-frozen fraction to vitrify and/or add cryoprotectants that are good glass formers to increase the glass transition temperature. DMSO is a broadly used permeating cryoprotectant which improves the post-thaw recovery of a broad range of cells and tissue types [40]. DMSO is a well-known "glass former" that works by increasing the total concentration of solutes in the system, reducing the total amount of ice formed at a given temperature, and raises the glass transition temperature [41]. In the present work, we show that with the addition of 10% DMSO to plasma and using controlled cooling/warming, we rescue cryopreserved samples from significant losses in RNA yield (Fig 5). While this approach is easy to implement, future experiments will nonetheless need to evaluate longer-term preservation durations. This is especially important since others have shown RNA degradation in 2 year old banked samples in the presence of DMSO [21]. Since the glass transition temperature of 10% DMSO is ~-120˚C, storage temperatures of -80˚C are suboptimal for long-term storage. Hence future experiments may test storage conditions in liquid nitrogen. In many cases, however, it may be preferred to store at -80˚C to overcome maintenance costs, the need for periodic refilling, and allow for shipment of frozen specimens on dry ice. As such, methods to increase the glass transition temperature at -80˚C would be of great benefit. Examples of such compounds include disaccharides (sucrose, trehalose) and polymers (hydroxyethyl starch, polyvinyl pyrrolidone) [42,43].

In summary, we demonstrate the loss of EVs and RNA because of cryopreservation. Further, we demonstrate a cheap and easy method to improve the yield of RNA using the commonly used cryoprotectant agent, DMSO. While longer storage durations still need to be comprehensively studied, we postulate that frozen samples can nonetheless produce relatively high-quality RNA. However, future work is still required to improve methods of EV cryopreservation, handling, and downstream analysis of RNA as well as other bionalytes. In particular, future studies that spike in EV reference material in plasma and test different storage and thawing conditions would be an area of immediate interest.

## Supporting information

**S1 Fig. Impact of cryopreservation on particle rate of plasma-derived EVs.** All plasma samples were added directly to the -80˚C freezer (without a freezing container), thawed at room

temperature, and isolated using ultracentrifugation. (A-D) Particle rate expressed as line graphs of fresh versus stored samples are illustrated with a black or red line, respectively, for representative donors. A higher particle rate indicates samples have a higher concentration of EVs.
(DOCX)

**S2 Fig. Impact of cryopreservation on particle diameter and size distribution of plasma-derived EVs.** All plasma samples were added directly to the -80˚C freezer (without a freezing container), thawed at room temperature, and isolated using ultracentrifugation. (A-B) Representative histogram for two donors showing EV concentration binned by particle diameter (nm) with fresh samples shown in dark grey and stored samples in red.
(DOCX)

**S3 Fig. Impact of cryopreservation on RIN values of plasma-derived EVs.** All plasma samples were added directly to the -80˚C freezer (without a freezing container) and thawed at room temperature. All individual data points (or biological replicates) consist of paired fresh and stored samples that were obtained from the same individual in one blood draw. Solid lines between data points link the corresponding fresh and stored samples from a given individual. (A) RIN of EVs in stored versus fresh samples isolated using ultracentrifugation and extracted using the RNeasy spin columns are expressed as box-and-whiskers plots showing median, interquartile range, maxima/minima, and all individual data points (n = 10). (B-C) Representative RNeasy electropherograms showing RNA traces obtained from the Bioanalyzer. Fresh (black line) versus Stored (red line) denotes control (not frozen) versus samples frozen and thawed. (D) RIN of EVs in stored versus fresh samples isolated using ultracentrifugation and extracted using the miRNeasy spin columns are expressed as box-and-whiskers plots showing median, interquartile range, maxima/minima, and all individual data points (n = 9). (E-F) Representative miRNeasy electropherograms showing RNA traces obtained from the Bioanalyzer. Data were analyzed using the Wilcoxon signed-rank test ($p < 0.05$); an asterisk indicates statistical significance.
(DOCX)

**S4 Fig. RNase A treatment and immunoblots of EVs isolated from plasma and media.** (A) RNA Yield of EVs isolated from plasma using ultracentrifugation and treated with RNase A are expressed as box-and-whiskers plots showing median, interquartile range, maxima/minima, and all individual data points (n = 6). (B) RNA Yield of EVs isolated from media using qEV Columns and treated with RNase A are expressed as box-and-whiskers plots showing median, interquartile range, maxima/minima, and all individual data points (n = 4). (C) Positive immunoblot staining for CD9 for EVs isolated from plasma and media. Data were analyzed using the Mann-Whitney U-test ($p < 0.05$); an asterisk indicates statistical significance.
(DOCX)

**S1 Raw images.**
(PDF)

## Acknowledgments

We thank Octavio Hurtado and Laura Libby for management of the facilities and Lynne Stubblefield for administrative assistance. We also thank Drs. Charles Lai and Xandra Breakefield for supplying the TdTomato plasmid and Dr. Marsha Rich Rosner for supplying the MDA-MB-231 1833 (BM1) cells used in this study. Finally, we sincerely acknowledge and thank healthy volunteers who donated blood specimens.

## Author Contributions

**Conceptualization:** Shannon N. Tessier, Eduardo Reátegui, Daniel C. Rabe, Mehmet Toner, Shannon L. Stott.

**Data curation:** Shannon N. Tessier, Lauren D. Bookstaver, Cindy Angpraseuth, Cleo J. Stannard, Beatriz Marques, Berent Aldikacti, Daniel C. Rabe, Shannon L. Stott.

**Formal analysis:** Shannon N. Tessier, Lauren D. Bookstaver, Cleo J. Stannard, Beatriz Marques, Uyen K. Ho, Alona Muzikansky, Daniel C. Rabe, Shannon L. Stott.

**Funding acquisition:** Mehmet Toner, Shannon L. Stott.

**Investigation:** Shannon N. Tessier, Lauren D. Bookstaver, Cindy Angpraseuth, Cleo J. Stannard, Beatriz Marques, Uyen K. Ho, Berent Aldikacti, Daniel C. Rabe, Mehmet Toner, Shannon L. Stott.

**Methodology:** Shannon N. Tessier, Lauren D. Bookstaver, Cindy Angpraseuth, Cleo J. Stannard, Beatriz Marques, Alona Muzikansky, Berent Aldikacti, Eduardo Reátegui, Daniel C. Rabe, Mehmet Toner, Shannon L. Stott.

**Project administration:** Shannon N. Tessier, Shannon L. Stott.

**Resources:** Shannon N. Tessier, Mehmet Toner, Shannon L. Stott.

**Software:** Shannon L. Stott.

**Supervision:** Shannon N. Tessier, Eduardo Reátegui, Daniel C. Rabe, Mehmet Toner, Shannon L. Stott.

**Validation:** Shannon N. Tessier, Uyen K. Ho, Daniel C. Rabe, Shannon L. Stott.

**Visualization:** Shannon N. Tessier, Lauren D. Bookstaver, Daniel C. Rabe, Shannon L. Stott.

**Writing – original draft:** Shannon N. Tessier, Berent Aldikacti, Shannon L. Stott.

**Writing – review & editing:** Shannon N. Tessier, Uyen K. Ho, Alona Muzikansky, Daniel C. Rabe, Mehmet Toner, Shannon L. Stott.

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
