## [Decision Letter · Decision Letter 0]

24 Mar 2020

PONE-D-20-05469

Isolation of intact extracellular vesicles from cryopreserved samples.

PLOS ONE

Dear Professor Stott,

Thank you for submitting your manuscript to PLOS ONE. After careful consideration y two reviewers, we feel that your study has merit but does not fully meet PLOS ONE’s publication criteria as it currently stands. Therefore, we invite you to submit a revised version of the manuscript.

Specifically, concerns were raised regarding the experimental design and whether the results support the conclusions of the manuscript. Addressing the issue of whether the decrease in particle concentration is attributed to the loss of EVs rather than to the loss of other extracellular particles is of particular concern. In addition, one reviewer had difficulties in interpreting the figures and provided suggestions for improving them. Both reviewers also requested more information on how the experiments were conducted.

If you believe you can address all of the reviewer concerns, we would appreciate receiving your revised manuscript by May 08 2020 11:59PM. To enhance the reproducibility of your results, we recommend that if applicable you deposit your laboratory protocols in protocols.io, where a protocol can be assigned its own identifier (DOI) such that it can be cited independently in the future. For instructions see: http://journals.plos.org/plosone/s/submission-guidelines#loc-laboratory-protocols

We look forward to receiving your revised manuscript.

Kind regards,

Colin Johnson, Ph.D.

Academic Editor

PLOS ONE

Journal Requirements:

4. We note that you have a patent relating to material pertinent to this article. Please provide an amended statement of Competing Interests to declare this patent (with details including name and number), along with any other relevant declarations relating to employment, consultancy, patents, products in development or modified products etc. Please confirm that this does not alter your adherence to all PLOS ONE policies on sharing data and materials, as detailed online in our guide for authors http://journals.plos.org/plosone/s/competing-interests by including the following statement: "This does not alter our adherence to  PLOS ONE policies on sharing data and materials.” If there are restrictions on sharing of data and/or materials, please state these. Please note that we cannot proceed with consideration of your article until this information has been declared.

Reviewers' comments:

Reviewer's Responses to Questions

**Comments to the Author**

1. Is the manuscript technically sound, and do the data support the conclusions?

Reviewer #1: Yes

Reviewer #2: Partly

2. Has the statistical analysis been performed appropriately and rigorously? 

Reviewer #1: No

Reviewer #2: Yes

3. Have the authors made all data underlying the findings in their manuscript fully available?

Reviewer #1: Yes

Reviewer #2: Yes

4. Is the manuscript presented in an intelligible fashion and written in standard English?

Reviewer #1: Yes

Reviewer #2: Yes

5. Review Comments to the Author

Reviewer #1: Isolation of intact extracellular vesicles from cryopreserved samples

by Shannon N. Tessier , Lauren D. Bookstaver, Cindy Angpraseuth, Cleo J. Stannard, Beatriz Marques, Berent Aldikacti, Eduardo Reátegui, Mehmet Toner, and Shannon L. Stott.

This manuscript addresses the negative effects that cryopreservation may have on extracellular vesicles. The effect of two different cooling regimes, thawing regimes, storage periods, and RNA isolation kits, in addition of addition of the cryopreservation agent DMSO, on EV number, RNA yield and RNA quality was observed.

Overall the subject of the study is most interesting and the results should be useful for researchers in the EV field. I do have some concerns, however, that I think warrant a major revision of the manuscript. These comments apply in particular to the presentation of the results and the thoroughness of the Materials and Methods section.

Major concerns:

1. Figures

The figures are not all easy to interpret. For example, in Figure 2A, what is really Stored:Fresh? And what comprises one data point? Samples from the same donor? I think the presentation you use in Figure 2B should be included for all your figures, i.e., separate columns for fresh and stored samples. And it is quite unclear what the n value corresponds to. This should be described more clearly in the Materials and Methods section.

And in the manuscript text you write that the plasma-derived EV concentration of stored samples was only 38% of fresh, non-frozen samples. When I look at Figure 2C, it appears as though the particle concentration at all particle diameters is much lower than this. If the figure cannot be interpreted as such, please explain.

Additionally, you keep referring to D0 and D10 in the figure legends while these annotations are nowhere to be found (only in the supplementary figures).

You never explain to the reader what an electropherogram depicts (you mention it very briefly in the Discussion, but that is not sufficient).

In Figure 4, please chose a color other than blue. It is not that easy to distinguish it from the black line.

2. Materials and Methods

This section is generally lacking information. Please be more thorough, as suggested below:

-line 157: Add a comment that the method is described below

-line 158: Please explain the rationale for choosing 10-12 days. This amount of days seems in many respects meaningless low. In my own personal experience we normally have to store the EVs for much longer than that. I would think that holds true in many laboratories. While it certainly would have been impractical to wait for years (although I hope you stored some of your samples for a future study), you could have chosen the middle road and selected a few months.

-line 161: For how long were the samples kept in the freezing container?

-line 164: How many hours?

-line 167: How much 10% DMSO did you add?

-line 175, starting with “For ultracentrifugation,…”: Please revise the sentence; there is a grammar issue here.

-line 179: You mix verb tenses several places, including here. Stick to one tense.

-line 192: What is the rationale for choosing these two kits? Please explain here or in the Introduction.

-line 204: Again, I am having trouble deciphering the exact number of samples in your different experiments. With n=4 it is troublesome to perform a t-test. Additionally, did you check for normality?

Additional comments:

1. Check singular vs. plural verb forms, f.ex. in line 32.

2. Shouldn´t there be a space before listing the reference number in the text?

3. Line 66-73: Cumbersome language; please revise.

4. Line 109: Which are “all experimental conditions”?

5. Line 111 starting with “Further, …”: It is unclear what you mean.

6. Line 115: Specify that you mean drawing of blood.

7. Line 115: Please move “including liposomes”. As it stands, it reads that lipid bilayers are sensitive to liposomes.

8. Line 121: What do you mean by “biologics”?

9. Line 217: Please explain what particle rate signifies.

10. Line 236: This belongs in the Materials and Methods section.

11. Line 238, starting with “This is…”: This belongs in the Materials and Methods section.

12. There is no sense in reporting p-values to four decimals.

13. Line 247: Please explain the RIN value in the Material and Methods section. Not all readers will understand what it signifies.

14. Line 268: This belongs in the Materials and Methods section.

15. Line 271, starting with “Secondly, …”: A bit unclear. What do you mean by traverse in this respect?

16. Line 286: “bucket”?

17. Line 329: Please explain the issue of EV count vs. RNA yield a bit more closely.

18. Line 337: Please discuss if your results could have been affected by the rather harsh ultracentrifugation process. Ideally, you should have used an additional EV isolation method.

19. Line 357: This explanation should have been included in the Materials and Methods section. Same for line 363 (slow vs. fast).

20. Line 374: Sentence unclear. Please revise.

21. Line 387: Please explain glassy state/glass transition some more.

22. Line 397: What do you mean with “While longer storage durations are required”?

23. Line 457: The names of authors in reference 14 are all in capital letters.

Reviewer #2: Tessier et al. study the impact of a freeze thaw cycle of blood plasma on the integrity of EVs and their associated RNA cargo. Blood plasma was collected from healthy donors and stored at -80°C for 10-12 days in the presence or absence of cryoprotectant (DMSO) versus freshly processed. Stored samples were frozen directly or through means of freezing containers; and thawed at room temperature versus 37°C. EVs were separated from fresh or stored blood plasma by differential ultracentrifugation. Particle distributions were obtained by qNano measurements on EV preparations. In addition, RNA was isolated and yield was quantified. This study aims to contribute towards setting standards for blood processing for downstream EV analysis. Although of broad interest for the EV research field, there are several concerns.

1) Since the authors separate EVs from blood plasma by differential ultracentrifugation the EV preparations are contaminated with other extracellular particles including RNA containing ribonucleoprotein complexes and lipoprotein particles. qNano cannot distinguish different types of extracellular particles. Thus, from the qNano experiments it is impossible to conclude that the decrease in particle concentration is attributed to the loss of EVs rather than to the loss of other extracellular particles (or a combined effect).

2) The same holds true for the observed decrease in RNA yield. Is the decreased yield due to disruption of EVs or other extracellular particles? Did the authors measure RNA yield on total blood plasma, prior to EV separation? Would the authors observe the same losses if EV preparations from both fresh blood plasma and cryopreserved blood plasma were treated with a combination or not of RNase, protease and detergent?

3) Comment 1 and 2 also account for testing the use of freezing containers, thawing at 37°C and adding DMSO to blood plasma prior to storage.

4) From the experiments presented in the manuscript it cannot be concluded that the observed decrease in number of particles and RNA yield can be attributed to EVs. Additional experiments should be performed including:

-assessing RNA yield in total blood plasma (prior to EV separation)

-assessing RNA and particle yield after RNase and/or protease and/or detergent treatment

-performing further characterization of EV preparations by electron microscopy and protein analysis (EV and non-EV associated proteins) to assess the composition of EV preparations (cfr MISEV2018 guidelines)

5) An alternative strategy that can be implemented is by the spiking an EV reference material, as is recently increasingly reported in literature, in blood plasma prior to testing different storage and thawing conditions. If not possible to include in the current experimental design of the study, it can be discussed by the authors.

6. PLOS authors have the option to publish the peer review history of their article (what does this mean?). If published, this will include your full peer review and any attached files.

Reviewer #1: No

Reviewer #2: No

---

## [Author Response · Author response to Decision Letter 0]

10 Feb 2021

Detailed responses are provided in the attached Response to Reviewers document.

---

## [Decision Letter · Decision Letter 1]

9 Mar 2021

PONE-D-20-05469R1

Isolation of intact extracellular vesicles from cryopreserved samples.

PLOS ONE

Dear Dr. Stott,

I am pleased to state that both reviewers found the revised manuscript significantly improved. One reviewer listed a few minor changes to the text which would remove the ambiguity to lines 43, 81, and 157.

We look forward to receiving your revised manuscript.

Kind regards,

Colin Johnson, Ph.D.

Academic Editor

PLOS ONE

Journal Requirements:

Reviewers' comments:

Reviewer's Responses to Questions

**Comments to the Author**

1. If the authors have adequately addressed your comments raised in a previous round of review and you feel that this manuscript is now acceptable for publication, you may indicate that here to bypass the “Comments to the Author” section, enter your conflict of interest statement in the “Confidential to Editor” section, and submit your "Accept" recommendation.

Reviewer #1: (No Response)

Reviewer #2: All comments have been addressed

2. Is the manuscript technically sound, and do the data support the conclusions?

Reviewer #1: Yes

Reviewer #2: Yes

3. Has the statistical analysis been performed appropriately and rigorously? 

Reviewer #1: Yes

Reviewer #2: Yes

4. Have the authors made all data underlying the findings in their manuscript fully available?

Reviewer #1: Yes

Reviewer #2: Yes

5. Is the manuscript presented in an intelligible fashion and written in standard English?

Reviewer #1: Yes

Reviewer #2: Yes

6. Review Comments to the Author

Reviewer #1: Please see attached document for my specific comments for this manuscript, I recomennd minor revision at this point

Reviewer #2: (No Response)

7. PLOS authors have the option to publish the peer review history of their article (what does this mean?). If published, this will include your full peer review and any attached files.

Reviewer #1: No

Reviewer #2: No

---

## [Author Response · Author response to Decision Letter 1]

15 Apr 2021

Reviewer #1: Response to the Reviewer 

We thank the reviewer for careful consideration and additional comments to further improve the manuscript. We address your specific comments below. 

Question 1: Line 43: More common to write this as 47-52%.

Response 1: We have modified line 43 to read 47-52%, instead of 52-47%. 

Question 2: Line 81/112: Specify which media you refer to (cell culture supernatants? Or does it refer to the media in the section starting with 157?).

Response 2: We have changed line 81 and 112 to read “cell culture supernatants,” instead of media. 

Question 3: Line 157: This section is missing some information (BM1 palm-tdTomato cells? Not easy to understand for those who are not used to working with your material). 

Response 3: We have added information about the BM1- palm-tdTomato cells to the Materials and Methods section as well as included references which detail how they were generated. These changes are as follows: “BM1 cells are a highly invasive bone-metastatic variant of MDA-MB-231 cells (the most used cell line to study triple negative breast cancer) that generate a lot of EVs. To generate fluorescent EV reporters for direct visualization of cargo, a palmitoylation signal was genetically fused to the N-terminus of tdTomato [32,33].” 

Question 4: Could storage in a DMSO-containing solution impact later downstream analysis of f.ex. EV content? I believe the presence of DMSO can change fluxes of ions and molecules through cellular plasma membranes.

Response 4: this is a great question from the reviewer that is certainly important. DMSO should be removed during EV isolation, including ultracentrifugation and size exclusion chromatography, like the IZON qEV columns used in the present manuscript. Other common methods not used in the present manuscript to isolate EVs, including polymer-based precipitations, density gradient centrifugation and microfluidics isolation technology, should also remove DMSO during EV isolation. We have modified this sentence in the discussion to make this point: “Further, we describe how the addition of the cryoprotectant agent, DMSO, can overcome some forms of cryoinjury, although longer-term preservation studies are a necessary next step. Importantly, this method is relatively easy, does not require any expensive equipment, and is compatible with downstream assays since DMSO should be removed during isolation, making it feasible for research and clinical implementation.”

---

## [Decision Letter · Decision Letter 2]

26 Apr 2021

Isolation of intact extracellular vesicles from cryopreserved samples.

PONE-D-20-05469R2

Dear Dr. Stott,

We’re pleased to inform you that your manuscript has been judged scientifically suitable for publication and will be formally accepted for publication once it meets all outstanding technical requirements.

Kind regards,

Colin Johnson, Ph.D.

Academic Editor

PLOS ONE

Additional Editor Comments (optional):

Reviewers' comments:

Reviewer's Responses to Questions

**Comments to the Author**

1. If the authors have adequately addressed your comments raised in a previous round of review and you feel that this manuscript is now acceptable for publication, you may indicate that here to bypass the “Comments to the Author” section, enter your conflict of interest statement in the “Confidential to Editor” section, and submit your "Accept" recommendation.

Reviewer #1: All comments have been addressed

2. Is the manuscript technically sound, and do the data support the conclusions?

Reviewer #1: (No Response)

3. Has the statistical analysis been performed appropriately and rigorously? 

Reviewer #1: (No Response)

4. Have the authors made all data underlying the findings in their manuscript fully available?

Reviewer #1: (No Response)

5. Is the manuscript presented in an intelligible fashion and written in standard English?

Reviewer #1: (No Response)

6. Review Comments to the Author

Reviewer #1: (No Response)

7. PLOS authors have the option to publish the peer review history of their article (what does this mean?). If published, this will include your full peer review and any attached files.

Reviewer #1: No

---

## [Editor Report · Acceptance letter]

5 May 2021

PONE-D-20-05469R2 

Isolation of intact extracellular vesicles from cryopreserved samples. 

Dear Dr. Stott:

I'm pleased to inform you that your manuscript has been deemed suitable for publication in PLOS ONE. Congratulations! Your manuscript is now with our production department. 

Kind regards, 

on behalf of

Dr. Colin Johnson 

Academic Editor

PLOS ONE